# Parenteral Nutrition Overview

**DOI:** 10.3390/nu14214480

**Published:** 2022-10-25

**Authors:** David Berlana

**Affiliations:** 1Pharmacy Department, Vall Hebron Barcelona Campus Hospital, 08035 Barcelona, Spain; david.berlana@vallhebron.cat; 2Pharmacology, Toxicology and Therapeutic Chemistry Department, Faculty of Pharmacy and Food Sciences, University of Barcelona, 08028 Barcelona, Spain

**Keywords:** parenteral nutrition, adults, hospitalized patients, refeeding, malnutrition, macronutrients, lipids, parenteral nutrition associated complications

## Abstract

Parenteral nutrition (PN) is a life-saving intervention for patients where oral or enteral nutrition (EN) cannot be achieved or is not acceptable. The essential components of PN are carbohydrates, lipids, amino acids, vitamins, trace elements, electrolytes and water. PN should be provided via a central line because of its hypertonicity. However, peripheral PN (with lower nutrient content and larger volume) can be administered via an appropriate non-central line. There are alternatives for the compounding process also, including hospital pharmacy compounded bags and commercial multichamber bags. PN is a costly therapy and has been associated with complications. Metabolic complications related to macro and micronutrient disturbances, such as hyperglycemia, hypertriglyceridemia, and electrolyte imbalance, may occur at any time during PN therapy, as well as infectious complications, mostly related to venous access. Long-term complications, such as hepatobiliary and bone disease are associated with longer PN therapy and home-PN. To prevent and mitigate potential complications, the optimal monitoring and early management of imbalances is required. PN should be prescribed for malnourished patients or high-risk patients with malnutrition where the feasibility of full EN is in question. Several factors should be considered when providing PN, including timing of initiation, clinical status, and risk of complications.

## 1. Introduction

Successful intravenous nutrient administration by Dudrick and colleagues marked a major advancement in providing nutrition to patients unable to be fed orally or enterally, leading to the birth of parenteral nutrition (PN) [1]. Since then, PN has been widely used in pediatric and adult patients whenever oral or enteral nutrition (EN) is not possible, insufficient or contraindicated. Nevertheless, PN is a costly therapy and can also be associated with hepatobiliary, infectious and mechanical complications, as well as hyperglycemia, hypertriglyceridemia, and electrolyte disturbances [2,3]. A PN admixture contains several active ingredients, meaning that its prescription is one of the most complex routinely used ingredients in the hospital setting. Moreover, PN is considered a high-alert medication, thus its use requires policies, systems and practices focused on safety to minimize patient risk.

The aim of this review is to offer an overview of the use of PN focusing on PN indications, complications, formulations and its use in hospitalized adult patients, as part of a special issue “Nutritional support in digestive diseases, and nutritional implications of dietary interventions”. It is a summary of the indications for and complications of PN, as well as recommended formulations, PNs use for special patient groups, current guidelines and recently available evidence. The recommendations, guidelines, and advice proposed here are not the result of a systematic literature review but of expert judgment based on a review of the literature to advise on best practice. No formal rating of the quality of evidence or strength of recommendations was performed. This overview was conducted by searching available scientific databases for articles reporting on PN in hospitalized adult patients. To fully investigate the use of PN in hospitalized adult patients, an exhaustive search for eligible studies was performed in PubMed, Embase, Medline, Cochrane library and Web of Science databases using the appropriate terms up to September 2022. Further articles were obtained from the references of searched articles and general reading. As the topic “parenteral nutrition” is quite broad, some aspects are covered by a brief overview, with references given to provide greater detail. Practical recommendations and key points are provided as a summary at the end of each section. They represent the author’s opinion to help the reader focus on the most important issues and to highlight potential future research linked to each section.

## 2. Indications

PN is indicated in situations where enteral or oral nutrition is not possible or is insufficient. European and American Guidelines recommend EN in preference to PN in patients with a functional and accessible gastrointestinal tract when nutritional support is required [4,5,6]. The main indications for PN in adult subjects are intestinal failure (IF) due to disease or treatment (short bowel syndrome, inflammatory bowel diseases, intestinal pseudo-obstruction, radiation enteritis), high-output fistulas, severe intestinal obstruction or an inaccessible gastrointestinal tract.

IF has been defined as a reduction in gut functions below the minimum necessary for the absorption of nutrients from the gastrointestinal tract to maintain health and growth [7,8]. IF resulting in the under-absorption of macronutrients and/or water and electrolytes may require parenteral supplementation. IF has also been subclassified, based on onset and functional classifications, with type I (acute) patients typically receiving short term PN in hospital. Meanwhile, type II (prolonged acute) and type III (chronic) patients require PN over a course of weeks to months or months to years, respectively, and typically receive PN at home (especially type III patients) [4,7]. In fact, the most common underlying diseases or clinical statuses in chronic IF are also the most common indications for PN, such as mesenteric ischemia, surgical complications, intestinal pseudo-obstruction, and radiation enteritis.

Since malnutrition is associated with poor outcomes, nutritional support is indicated in patients who are malnourished or at risk of developing malnutrition [8]. In these patients with a gastrointestinal tract that is not functional or that cannot or should not be accessed, PN is indicated. A high prevalence of malnutrition has been found in hospitalized patients, being common in critically ill, surgical, and/or cancer patients. Examples of clinical conditions requiring PN are listed in Table 1.

Proper use of PN maximizes its clinical benefit while minimizing the potential risk of complications. The indication and aims of PN should be identified clearly at the beginning and revised frequently, so as to verify if an enteral route may be available and to ensure adequate nutrition provision while minimizing the risk of complications. However, PN use should not be based solely on medical diagnosis or disease state. Before initiating PN, a full assessment of the feasibility of EN should be conducted, so that PN is reserved for clinical conditions where adequate EN is not an option. Nevertheless, there are also clinical statuses where patients with oral nutrition or EN receive less than 60% of what they require. In these situations, supplemental PN is also indicated to achieve estimated nutritional requirements. A typical situation is a patient with chronic IF unable to absorb enough nutrients and receiving supplemental home PN for less than 7 days per week.

The timing of PN initiation is another key question, but studies show contradictory results. However, clinical guidelines recommend beginning PN (supplemental or full) in well-nourished patients who are not able to achieve necessary nutritional requirements after 7 days [5,9,10]. In patients at risk of malnutrition, clinical guidelines recommend starting PN earlier, within 3–6 days, if they are unlikely to achieve satisfactory oral nutrition or EN. Nevertheless, in malnourished patients, beginning PN as soon as is feasible in those for whom oral nutrition or EN is not sufficient or possible is also recommended.

The implementation of nutrition support teams has been shown to reduce the inappropriate use of PN, leading to a reduction in complications and costs related to PN [11,12,13,14,15]. The common cut-off point of less than 7 days of PN to define inappropriate PN in these studies may lead to some controversy and a reconsideration of the results. However, guidelines recommend PN initiation only if the duration is anticipated to be more than 7 days. The inappropriate use of PN increases the risk of complications, leading to an increase in hospitalization and morbidity, as has been shown in the EPANIC study [16]. Reanalysis of this trial has demonstrated significant adverse effects with the use of PN outside of intestinal failure, suggesting the need for strict assessment of the use of PN in elective surgery.

Practical recommendations/key points:Hospitalized patients should be regularly screened for risk of malnutrition, especially those who might be candidates for PN.PN is not an emergency treatment and should be started electively and revised frequently to check if the enteral route might be available.Developing support policies and procedures is also recommended to assist with the decision-making for PN initiation, as well as implementing a quality improvement process to ensure appropriate use of PN.Future research: Optimal timing of PN and its relationship with clinical and nutritional outcomes and costs, involving a homogenous patient population.

## 3. Vascular Access and Administration

### 3.1. Vascular Access

Since many of the complications associated with PN are linked to the presence of a vascular access device (VAD), the proper selection and placement of the VAD is essential to avoid or minimize such complications. By using a central VAD, many of the technical problems inherent in peripheral PN (PPN) can be avoided. Nevertheless, there are still adverse events related to central PN administration [3,8].

Placing a catheter, whether centrally or peripherally, has inherent risk that can appear immediately during placement or post-procedure. Although peripheral VADs may be suitable for administering a dilute nutrient admixture, they also have a high rate of technical failure. The adequacy of the vein limits the use of the peripheral system for such infusions, since most of the PPN formulations are between 750 and 900 mOsm/L [6,17,18]. These formulations are based on a decreased dextrose concentration and osmolarity (by increasing final volume). Therefore, patients with fluid restriction should not be candidates for PPN due to the risk of fluid overload to achieve their energy requirements. Peripheral PN is recommended for short-term therapies (≤10–14 days) because of the low reliability of peripheral VADs, frequently cited as the cause of phlebitis and extravasation. Moreover, because of the lower dextrose and protein content, there is less chance of achieving adequate nutritional requirements with PPN. Thus, it might be used supplementally or as a bridge nutrition therapy during transition periods where the clinical status of the patient does not justify inserting a central VAD [18,19].

Midline catheters are VADs that are inserted in the peripheral veins of the upper extremities with the tip located around the axilla. They avoid the central venous veins, are longer and sit deeper than traditional peripheral catheters, and are a feasible alternative for peripheral PN. As deeper veins may support solutions with higher osmolarity, lower rates of phlebitis may be expected with midline catheters compared to conventional peripheral VADs. However, midline VADs are still prone to phlebitis, and their deeper location may mask relevant signs and symptoms of phlebitis [3,20]. Age may also play a role in the development of phlebitis, and adults, especially elderly patients, are more likely to develop phlebitis compared to pediatric patients [21,22]. Thus, it is reasonable to limit peripheral PN < 900 mOsm/L for pediatric patients and <800–850 mOsm/L for adults [3,6,22,23].

For administration of hyperosmolar PN admixtures, a central VAD is needed. The catheter tip should be placed in the distal vena cava or right atrium. The most common insertion sites include the subclavian, internal jugular, femoral, cephalic and basilic veins. Although the femoral vein can be used, it is associated with a greater risk of catheter-related bloodstream infection (CRBSI) and thrombosis [3,24,25]. The use of central VADs allows for the administration of solutions not limited by pH, osmolarity or volume. Central VADs can be grouped into the following four categories: peripherally inserted central catheters (PICCs), nontunneled (inserted into jugular, subclavian or femoral vein), tunneled and implanted [3,8,24,26]. The correct positioning of all newly inserted central VADs should be confirmed before PN administration, either radiographically, fluoroscopically or by using ultrasonography guidance during insertion. Today, a common and easy way to insert PICCs at the bedside is by using ultrasound and ECG guidance. In fact, the use of ultrasound for VAD placement is strongly recommended to reduce the number of complications and to increase the safety and quality of the VAD placement [27].

Several factors should be considered when a central VAD is selected for PN: risk of infection (mainly CRBSI), the patient’s medical condition, concurrent intravenous therapies, anticipated PN duration, the setting of PN administration, and the complexity of VAD care. Table 2 summarizes the type of VADs used for PN administration.

### 3.2. Administration

Patients who require short-term PN, representing the majority of hospitalized patients with PN, typically receive PN as a 24 h continuous infusion. Administration over 24h enables less manipulation and a lower infusion rate, limiting the overloading of glucose as well as fluids. However, home PN is often administered on a cyclic (discontinuous) schedule. Cyclic administration during a portion of the day or night allows the patient freedom from the intravenous tubing and pump apparatus [4,7]. Cyclic PN administration has also been used as a strategy against liver impairment associated with PN [28,29]. When cyclic administration is proposed, glycemia should be monitored to avoid hypoglycemia after discontinuation, as well as hyperglycemia due to the increased rate of infusion.

Since there is a significant risk of infusing particulates as well as precipitates from single elements of the mixture, recommendations to use in-line filters during PN have been made in the United States and some countries in Europe [30,31,32]. Current recommendations to reduce this risk include filter use during PN administration, particularly for those patients with the highest risk of prejudicial effects (e.g., critically ill, immunocompromised, neonates). In fact, pediatric guidelines, with a strong consensus, recommend the use of in-line filters in pediatric PN [33]. A 1.2-micron in-line filter is considered appropriate for lipid-containing admixture, but a 0.22-micron filter could be used for non-lipid-containing admixtures [32]. Although, when the lipid is administered separately, a single container infused over a maximum of 12 h is recommended to reduce the risk of contamination and infection; lipid administered separately is commonly infused over 24 h [34,35]. However, the separate administration of the lipid may lead to multiple manipulations, increasing the risk of catheter-related infection and cost [36].

During storage and administration, PN admixtures should be protected from light due to the photodegradation of some nutrients [6,37,38,39]. In fact, there is general agreement to recommend photoprotection in PN for the pediatric population due to toxic degradation of PN ingredients (mainly lipids and vitamins) linked to adverse effects [40,41,42]. However, this recommendation should also be extended to PN for adults, since components used for adults are also susceptible to photo-oxidation. Moreover, the use of multilayered bags is also recommended to prevent oxidation [37,39,43].

Practical recommendations/key points:Protocolize VAD placement, individualizing the selection of VADs based on risk and benefits, and clinical factors, and validate the optimal position of the VAD before PN initiation.Use a proper in line filter for administration and avoid additional lines for lipid administration.Future research: Developing VADs and administration kits with easier and safer procedures. Evaluating the effects of accumulation of particles in vital organs and the interest of in-line filters.

## 4. Composition of PN Admixtures

Once PN is indicated, a mixed nutritional solution (carbohydrates, protein, and lipids) is recommended to avoid nutrient deficiencies. PN components include fluids, macronutrients (amino acids, carbohydrates, and lipid emulsions) and micronutrients (electrolytes, vitamins, and trace elements). The PN prescriber should be well versed in the appropriate indications for PN, the vascular access devices (peripheral and central) and their associated complications, as well as the appropriate amounts of each macronutrient and micronutrient to be included in the PN. Historically, PN was administered in separate bottles containing a carbohydrate solution, an amino acid hydrolysate, and an intravenous fat emulsion (IVFE). However, this system has been associated with higher costs and risks of infection, since it requires multiple IV lines [44]. 

Other systems that can be used for PN administration include a two-in-one system (containing amino acids and glucose) or an all-in-one system (containing amino acids, fat and carbohydrates). The two-in-one system includes amino acids and glucose in a single bag along with micronutrients but requires separate administration of the lipid product. In an all-in-one system, also called total PN (TPN) or total parenteral admixture (TNA), all nutrients are mixed in a single bag and infused simultaneously. As PN can be used supplementally when oral nutrition or EN is not sufficient, full PN is also called TPN to differentiate it from supplemental PN.

Moreover, with regard to the composition of PN, full PN (TPN) should be differentiated from PPN, since PPN has limited macronutrient content to reduce the total osmolarity of the mixture. To reduce osmolarity, PPN should have an increased total volume as well as reduced solutes (mainly macronutrients). The estimated contribution to osmolarity, measured as mOsm per gram of amino acid, dextrose and intravenous lipid emulsion (IVLE) 20%, is 10, 5, and 0.7, respectively [45]. General recommendations for PPN macronutrient content are as follows: amino acids < 4%, glucose < 10%. Consequently, a lower load of amino acids and glucose is often compensated for by an increased lipid load to achieve caloric goals due to the lower osmolarity of the lipids.

As midline catheters are becoming an increasingly popular way to administer PN in the hospital setting, the use of PPN is also growing, particularly in acute pancreatitis (AP), inflammatory bowel disease (IBD) and surgical patients [21,46,47,48]. In fact, PPN is largely used in the perioperative period for selected patients, as it has been included in some perioperative procedures [47,49]. In fact, PPN is currently used for a wide variety of clinical and logistical reasons [21,46]. However, the disadvantages of PPN, such as low concentrations of macronutrients, large fluid volumes, and the risk of phlebitis, may limit the provision of adequate nutrition.

There are two choices for the compounding process, namely hospital pharmacy compounded bags (HPCBs) and commercial multichamber bags (MCBs). HPCBs must be prepared almost daily by the pharmacy mainly because of the need for customized PN to cover individual needs, especially in critically ill patients, but also because of limited mixture stability, thus, special equipment and infrastructure are needed. HPCBs can be tailored to achieve the individual needs of a patient or can be prepared as a standard PN to cover the needs of a group of patients. Moreover, outsourcing the preparation of PN is a service offered by some companies available to many hospitals. The main use of this service is to provide PN to home PN patients, but also for inpatients, by compounding and delivering the PN; adjusting the shipment depending on the needs of the patient as well as the stability of the admixture once or twice per week. On the contrary, the shelf-life of most commercially available MCBs is more than 24 months at room temperature. MCBs contain a fixed number of macronutrients with or without electrolytes in separate compartments within a single bag; nutrients are mixed during preparation by breaking the plastic seals between compartments. Vitamins, trace elements, and electrolytes (if needed) are added to the bag during the compounding process prior to dispensing for administration. However, MCBs have advantages over HPCBs and multi bottle systems, including reduced costs, time and labor, and fewer errors in PN preparation [44,50,51], as well as their stability being guaranteed by the manufacturing company. In addition, electrolytes and nutrients, such as glutamine or omega-3 fatty acids, can be added to MCBs because these criteria have been considered in the development of MCB formulations. In contrast, by prescribing and compounding a tailored PN, it has been claimed that final macronutrient concentrations should be as follows: amino acid ≥ 4%, dextrose ≥ 10%, and lipids ≥ 2% to keep the admixture stable [37]. However, these recommendations do not agree with macronutrient concentrations recommended for PPN formulations. In fact, MCB formulations for PPN with lower concentrations of amino acids (<3%) and dextrose (<8%) have been shown to be stable for up to 7 days according to the manufacturing company.

The specific doses and combinations of nutrients can greatly affect the stability of a PN admixture. Even micronutrients such as electrolytes, vitamins and trace elements can affect PN stability, especially in an all-in-one PN. Different factors are known to affect the compatibility of any given additive including temperature, pH, concentration (even trace elements have been involved in the formation of precipitates), order of mixing and length of time of exposure [37,45,52,53,54,55,56,57,58]. The most common compatibility issues are as follows:Calcium and phosphorus: Solid precipitates may develop with the addition of an incompatible combination of salts to a PN mixture. Different factors affect the formation of these precipitates, such as temperature, time of exposure, pH and order of mixing the PN ingredients. Organic salts, such as calcium gluconate (for calcium) and sodium glycerophosphate (for phosphorus), are more stable than the equivalent inorganic salts.Bicarbonate salts should be avoided due to incompatibility with PN admixtures, if needed, acetate salts can be added.Medication additives: Despite there being data about the compatibility of several drugs with PN, PN should not be used as a drug delivery vehicle.

With regard to the stability of the admixture, some nutrients have a destabilizing effect in the presence of other nutrients. Similarly, there are general factors affecting PN admixture stability [6,35,37,38,39,43,45,55,58,59,60,61,62,63,64]:Temperature: For example, with increased temperature, there is a raised likelihood of calcium-phosphate precipitate formation and of the degradation of several amino acids.pH: The optimal pH for IVFE stability is in the range of 6–9, whereas calcium phosphate solubility is better at lower pH values.Light and oxygen exposure: Enhances the degradation of some vitamins and amino acids.All-in-one mixtures: All the factors that affect stability in a two-in-one formulation tend to do so to a greater extent with an all-in-one mixture. Therefore, a more conservative approach to interpreting guidelines for compatibility is needed to prevent the destabilization of the TNA or formation of a precipitate. The physical instability of IVLEs refers to lipid droplet size (e.g., a high concentration of cations may lead to destabilization of the emulsion and limit the load of cations that may be administered in the TNAs).Vitamins: Light exposure and time are the common factors affecting vitamin stability.Composition, due to compatibility issues.

Both PN prescribing and compounding methods, standardized by using MCBs and customized PN, should be based on the patient’s clinical condition. The pharmacist has a responsibility to strictly follow all guidelines, standards, validated methods, and recommendations related to compounding and delivery of safe PN; this pertains regardless of the use of an automated compounding device, manual compounding or the use of MCBs [6,36]. Standard ranges for protein and energy requirements incorporate dosing weight, adjusted for obesity. The usual distribution of nonprotein calories is 50% to 80% as carbohydrates and 50% to 20% as lipids [8,45]. In the case of supplemental PN or a combination of oral, EN and PN, all energy and protein intakes should be taken into account.

Practical recommendations/key points:Develop and maintain safety protocols for all PN processes, from prescribing and compounding steps to final administration.For stability and compatibility reasons, additions of high loads of electrolytes in the PN admixture should be avoided, especially cations.Future research: Stability studies to assess compatibility and stability of PN and PN components.

### 4.1. Proteins

In patients receiving PN, protein needs are met by supplying intravenous, sterile, free amino acid solutions that also provide energy (4 kal/g) [8,45]. As a source of nitrogen, amino acids are used in protein synthesis and in replacing protein stores that have been depleted secondary to disease. Nitrogen content varies depending on the concentration of the amino acids; however, amino acid solutions are generally assumed to be 16% nitrogen (6.25 g of protein = 1 g of nitrogen) [8,65]. To optimize nitrogen retention, sufficient non-protein energy substrates must be provided to prevent oxidation of amino acids. Non-protein calories per nitrogen ratio range from 125 to 225 kcal/g N for non-stressed PN patients. According to the American Society for Parenteral and Enteral Nutrition (ASPEN) guidelines, the best non-protein calories/nitrogen ratio is from 70:1 to 100:1 for critically ill patients, reduced to 30:1 to 50:1 for obese critically ill patients [65,66].

Commercial amino acid solutions for PN contain essential amino acids and non-essential amino acids to reach the required quantity of nitrogen. Although commercially standard amino acid solutions for PN have a different ratio of essential to non-essential amino acid, all the amino acid mixtures meet the essential amino acid dose at usual doses; in general, the usual dose of protein is 1 g/kg of body weight for unstressed healthy patients [65]. The recommended dose of protein in patients with acute renal failure is 0.6–0.8 g/kg/day without renal replacement therapy and 0.8–1 g/kg/day in chronic renal failure. The recommended dose is increased when renal replacement therapy is implemented (1.2–1.5 g/kg/day with hemodialysis and 1.3–2 g/kg/day with continuous renal replacement therapy) [67,68]. The use of standard amino acid PN formulations in acute kidney injury is not recommended since there is no evidence at this time to support the use of special PN formulations in acute renal impairment.

An amino acid mixture containing large amounts of branched-chain amino acids (BCAAs) and lower doses of aromatic amino acids (phenylalanine and tyrosine), methionine, and tryptophan are also commercially available. This mixture has been suggested as a liver-adapted formulation to be used for grade III to IV hepatic encephalopathies [69]. However, according to several guidelines, there is no evidence to suggest that BCAA-enriched formulations improve outcomes compared with standard amino acid mixtures in critically ill patients with PN and liver disease [66,70]. Nevertheless, the current European Society for Clinical Nutrition and Metabolism (ESPEN) guidelines recommend BCAA-enriched formulas in patients with hepatic encephalopathy in need of EN [70].

### 4.2. Carbohydrates

The most commonly used carbohydrate substrate is dextrose, which in its hydrated form provides 3.4 kcal/g carbohydrate (mainly used in the USA and Canada). Meanwhile, a non-hydrated form of dextrose, mainly used in Europe, provides 4 kcal/g carbohydrate [8,45]. Carbohydrates usually provide 50–60% of total daily calories. Decreased glucose utilization occurs with advancing age, liver disease, sepsis, stress (such as trauma, burns or surgery), and medications (such as corticoids, tacrolimus). Based on dietary reference intakes, a minimum of 130 g of carbohydrates must be provided in healthy patients. Continuous dextrose infusion rates in adult patients should be kept at ≤4–7 mg/kg/min to avoid hyperglycemic episodes if the oxidation glucose rate is overtaken [6,8].

### 4.3. Lipids

Lipid emulsions are an important component of PN because they supply essential fatty acids and minimize the dependence on glucose as a major source of non-protein energy. Commercially available IVFEs for PN have a 20% concentration. Each gram of fat provides 9 kcal; however, the glycerol in IVFE adds calories such that each gram of fat in IVFE 20% is equivalent to 10 kcal [34,71]. Lipids usually provide 20–30% of total daily calories; higher amounts of lipids might lead to hypertriglyceridemia and fat overload syndrome. General recommendations are to have IVFE doses of no greater than 1 g/kg body weight/day to avoid lipid overload [8,29,72].

The first commercial fat emulsion was based on a soybean oil-based lipid mixed with egg yolk [73]. The currently available IVFEs are derived from soybean, safflower, coconut, olive or fish oil. Soybean-based IVFEs have a high concentration of polyunsaturated fatty acids (PUFA), containing essential fatty acids, and are also high in phytosterols [71,74,75]. Coconut oil is the source of medium-chain triglycerides (MCT) in IVFE. However, MCTs do not contain any essential fatty acids. Similarly, olive oils have a low essential fatty acid content; thus, blending with a soybean base is necessary to prevent fatty acid deficiency. Fish oil contains omega-3 fatty acids and alpha-tocopherol, but no essential fatty acids.

The biological effects of lipid emulsions are strongly influenced by their fatty-acid composition [74]. Lipid emulsions containing only soybean oil have high concentrations of the omega-6 PUFA linoleic acid, which is transformed to arachidonic acid (AA), a precursor to eicosanoids that promote inflammation and suppress cell-mediated immunity [76]. Medium-chain triglycerides (derived from coconut oil or palm oil) and olive oil are considered to be less inflammatory than soybean oil, while lipid emulsions containing fish oil are rich in omega-3 PUFAs, such as docosahexaenoic acid (DHA) and eicosapentaenoic acid (EPA), which exhibit anti-inflammatory, immunomodulatory, and antioxidative properties in preclinical models [74,76,77]. The omega-3 PUFAs found in fish oil, mainly EPA and DHA, compete with AA (an omega-6 PUFA) for the lipoxygenase and cyclooxygenase pathways for the synthesis of eicosanoids, which are lipid mediators typically involved in inflammation activation [76,78]. Thus, fish oil has anti-inflammatory potential due to interfering with the AA pathway and by producing less proinflammatory eicosanoids. There are clinical data suggesting that omega-3 PUFAs have beneficial effects on the immune system and organ function and improve clinical outcomes in surgical and acute respiratory distress syndrome [79,80]. The use of pure fish oil or fish-oil enriched IVFEs has been related to a lower risk of hepatic dysfunction, as well as the recovery of liver abnormalities [81,82]. A lower phytosterol content may play an important role in this, since phytosterols have been linked to a reduction in bile flow [83]. Characteristics of IVFEs currently available in Europe are shown in Table 3.

### 4.4. Micronutrients: Electrolytes, Vitamins and Trace Elements

Standard ranges for electrolytes and trace elements are age-specific and based on normal organ function and normal losses. However, the electrolyte load may also be limited due to compatibility issues [37]. Electrolyte and fluid requirements vary depending on the patient’s renal, fluid, and electrolyte status, as well as their underlying disease and losses [6,9,35]. Baseline serum electrolyte measurements are recommended before ordering a PN solution and then electrolytes are added accordingly. Medications can also influence electrolyte status and should be taken into account when ordering PN. The most common causes of electrolyte disturbance (excess) are impaired renal function (leading to hyperkalemia, hypermagnesemia and hyperphosphatemia), medication administration, or receiving too high an IV electrolyte load. Periodic monitoring of serum electrolytes is required, especially in critically ill and malnourished patients. According to current guidelines, vitamins and trace elements should be administered daily, and therefore routinely added to PN solutions, except in case of overload; high doses of micronutrients should not be administered without proven deficiency [86].

Practical recommendations/key points:Protein dose recommendations are still controversial; 1–1.2 g/kg/day is a reasonable dose with no renal impairment.Lipids based only on soybean oil contain high concentrations of PUFA, omega-6 fatty acids and phytosterols. Consequently, more recently developed lipid emulsions have partially replaced soybean oil with other oils (MCT, olive and fish oil), thereby reducing phytosterols and increasing omega-9 and omega-3 fatty acids.Monitoring of fluid and electrolytes losses, as well as renal function to control electrolytes and fluid imbalances, should be implemented in all patients receiving PN.Future research: Optimal protein dose with renal replacement therapy.

## 5. Complications and Monitoring

### 5.1. Complications

Parenteral nutrition-associated complications may be categorized as metabolic, infectious, and mechanical (mainly linked to VAD).

#### 5.1.1. Hyperglycemia

Hyperglycemia remains the most common complication of PN. A high prevalence of hyperglycemia during PN therapy has been reported even in non-critically ill patients [87,88]. Tight glycemic control of below 110mg/dL has been shown to reduce mortality and morbidity in critically ill patients [89]. However, it was also later shown that intensive glucose control increased hypoglycemic events and mortality, and that glycemic control close to 180 mg/dL resulted in better outcomes compared to lower targets [58,59]. Poor glycemic control in patients receiving PN has been associated with poor outcomes, both in critically ill and non-critically ill patients [66,87,88,90]. Preventing hyperglycemia in PN is possible by starting PN slowly and monitoring glycemia levels frequently. Avoiding high amounts and limiting glucose in PN, as well as adding insulin to PN are also strategies that might prevent hyperglycemic events in patients with PN [6,8,88,91]. Predictors found to be associated with hyperglycemia during PN therapy include amount of glucose administered, critically ill patients, age > 65 years, underlying diabetes, presence of infection, renal impairment and concomitant use of glucose-elevating drugs (e.g., glucocorticoids, tacrolimus, somatostatin or octreotide) [88,90,92]. Although hypoglycemia occurs less often than hyperglycemia, its consequences can be life threatening. Hypoglycemia may result from providing too much insulin or stopping a PN infusion abruptly.

Practical recommendations/Key points.

Monitoring glucose is mandatory in patients with PN, more frequently at the beginning of PN, reducing frequency after stabilization.Identifying risk factors and drugs that may lead to hyperglycemia.Future research: Combined or single insulin regimens with PN for glycemic control.

#### 5.1.2. Hypertriglyceridemia

Lipid overload has been associated with hypertriglyceridemia and liver dysfunction. A normal dose of IVFE is below 1.5 kg/kg/day, including lipids from non-nutritive sources such as propofol, but up to 1 g/kg/day has been recommended in patients at risk of hypertriglyceridemia [29,72]. Hypertriglyceridemia is a frequent metabolic complication associated with fat administration in PN. In general, hypertriglyceridemia can occur if the infusion rate of IVFE exceeds the capacity of plasma fat clearance, but also occurs with glucose overfeeding. Therefore, energy requirements should always be monitored and adjusted accordingly. However, hypertriglyceridemia has also been described in patients receiving a normal dose of IVFE. In stressed patients, as well as in renal impairment, lipoprotein lipase activity is decreased leading to an accumulation of lipids in blood [29,93]. Factors found to be associated with hypertriglyceridemia include sepsis, renal failure, hyperglycemia, obesity, alcoholism, pancreatitis, high-output fistula, multiple organ failure, pre-existing hyperlipidemia and co-administration of drugs such as corticosteroids, cyclosporine, tacrolimus, sirolimus or propofol [8,29,93].

Acceptable serum triglyceride concentrations for those receiving PN are <400 mg/dL [26]. In patients with triglyceride levels close to 400 mg/dl or higher, IVFE should be reduced or discontinued. Additionally, a lowering of dextrose has also been suggested if hypertriglyceridemia is thought to be associated with dextrose overfeeding. Reduction or withdrawal of the fat content also leads to a reduction of the energy provided by the PN [8,34]. IVFE containing fish oil and MCT might reduce the risk of hypertriglyceridemia by accelerating lipid clearance and this has been suggested as a means of dealing with hypertriglyceridemia whilst still maintaining energy intake [94].

Practical recommendations/Key points:Use of omega-3 fatty-acid enriched lipids and limiting lipid intake is recommended (<1 g/kg/day including external sources such as propofol) to avoid hypertriglyceridemia.To help monitoring, when hypertriglyceridemia is present, blood samples should be properly collected to avoid possible artifactual results.Future research: Exploring strategies such as omega-3 fatty acid dose levels and administration, to deal with and control blood triglyceride levels without a reduction of caloric intake because of lipid restriction.

#### 5.1.3. Refeeding Syndrome

Refeeding syndrome (RS) is a potentially fatal metabolic complication in severe cases and is characterized by fluid complications and electrolyte imbalances occurring during the beginning of nutritional support. RS is characterized by a reduction in serum levels of one or a combination of electrolytes. The most sensitive electrolyte is phosphorus, but RS may also present as a reduction in potassium and/or magnesium [95,96,97]. In addition, thiamine deficiency has also been described in RS. These reductions can develop in hours to days after first introducing energy to a patient who has been exposed to a substantial period of malnutrition. The main clinical symptoms associated with the syndrome are water retention, heart failure, pulmonary edema, neuromuscular paralysis and even death. Historically, reports of RS have focused on patients with eating disorders, adult patients malnourished because of underlying medical conditions or those with chronically decreased oral intake [95,96]. However, unexpectedly high incidences of RS have been reported in hospitalized patients, even with PN [98]. The ASPEN consensus recommendations for RS proposed the following criteria for stratifying patients as moderate and high risk for refeeding [95]:Low BMI < 18.5 kg/m^2^;Recent weight loss of 5% in 1 month or 7.5–10% in 3 to 6 months;None or negligible oral intake 5–6 days;Caloric intake < 75% estimated for >5 days during acute illness or injury;Caloric intake < 75% estimated energy for >1 month;Abnormal potassium, phosphorus, or magnesium serum concentrations;Loss of subcutaneous fat;Loss of muscle mass;Higher-risk comorbidities (diseases and clinical conditions associated with the presence of the prior criteria, such as alcoholism, eating disorders, cancer, malabsorptive states, etc.).

Depending on the severity of the criteria or presentation with at least one or two criteria, the patient would be classified as a significant or moderate risk for RS. To prevent RS, it is useful to screen at-risk patients before beginning nutritional support [95,96]. Water balance, cardiovascular function and serum electrolytes should be carefully monitored at the start and during PN. Serum electrolyte imbalance, particularly phosphorus and potassium, should be corrected before starting PN. Thiamine and B group vitamin replacement should also be considered. The main guidelines recommend starting with a low caloric intake (10–20 kcal/kg/day or less in extreme cases) in patients at risk of RS, increasing slowly over 4–7 days to meet requirements [95,96,97].

Practical recommendations/key points:Protocolize identification of risk factors for RS and patients at risk of RS.Future research: Monitoring and new approaches to measuring the risk of overfeeding. Compare the effectiveness for avoiding RS of different initiation regimens and protocols.

#### 5.1.4. Hepatobiliary Complications

Disorders of the liver and biliary system are complications commonly reported in patients receiving PN. Parenteral nutrition-associated liver disease (PNALD) is a spectrum of diseases that can range from mild liver enzyme abnormalities to steatosis to eventual fibrosis or cirrhosis [8,99]. There are three primary types of PNALD: steatosis, cholestasis, and gallbladder sludge/stones. Patients may have one of these disorders or a combination of the three. Other terms for PNALD, intestinal failure-associated liver disease (IFALD) and parenteral nutrition-associated cholestasis (PNAC), have been used interchangeably [99,100]. Steatosis is the accumulation of fat in the liver usually caused by providing excess energy in PN. This presentation typically occurs within 2 weeks of PN initiation. Cholestasis is caused by the impaired secretion of bile or a biliary obstruction. High conjugated bilirubin levels are the main indicator that cholestasis is present. Rather than being a direct result of PN infusion, the lack of enteral stimulation leads to impaired bile flow and gallbladder contractility, and consequently, gallbladder stasis and gallstone formation [99,100,101]. PNALD, or rather PNAC, is often defined biochemically as 1.5 times the upper limit of normal elevation of two out of the following liver test: gamma-glutamyl transferase or alkaline phosphatase and/or serum conjugated bilirubin ≥2 mg/dL [100,102]. Elevation occurs within 1 to 3 weeks of initiating PN [99,101,102]. Although changes typically seen in PNALD occur in long-term PN, biochemical signs may occur even after the first week.

The mechanisms of PNALD remain unclear but seem to be multifactorial. Some factors associated with its development are related to the nutrient composition of the PN admixture, whereas others are unrelated to PN use. The basal clinical state (liver or biliary disease, abdominal surgery and sepsis), numerous invasive procedures, infections and liver-insulting drugs may also be present with PN therapy and contribute to alterations in liver tests [8,103,104]. Abnormal liver tests are also seen in postoperative or critically ill patients who receive PN, frequently due to hypoxic liver injury or ischemic hepatitis. There are also factors inherent in PN itself and in the lack of enteral feeding. PN admixture may be toxic due to its components and calorie overload. Hepatic steatosis and hepatitis may occur in any case of overfeeding, regardless of the route of nutrient administration. Excessive carbohydrate or lipid administration can lead to steatosis [99,100]. IVFE can also have a significant effect on the development of PNALD. Soybean-oil based IVFE contains high doses of ω-6 fatty acids (proinflammatory fatty acids) and large amounts of phytosterol [74,84]. The combination of these omega-6 fatty acids and phytosterols can contribute to liver impairment, especially for patients receiving long-term PN therapy with soybean oil-based IVFE, leading to PNALD. Pure fish-oil-based IVFE has been shown to reverse PNALD in pediatric patients when used in place of a soybean-oil-based emulsion [81,82]. Besides the minimal content of phytosterols, this IVFE contains anti-inflammatory omega-3 fatty acids, which could potentially decrease the risk of PNALD development or provide treatment for those with preexisting PNALD.

There are recommendations, such as modification of the PN formula, that can reduce the risk of PNALD development. Preventing excess energy administration and providing an appropriate ratio of energy from glucose and fat is important in its prevention [8,29]. The early reintroduction of gut feeding for intestinal stimulation, as well as for lowering parenteral caloric intake, is also recommended to reduce the risk. Here, the cycling of TPN, given the periods of fasting, has been postulated to help with fat mobilization and reduction of hepatic steatosis [28,105]. The use of new generation IVFEs with a lower content of soybean oil, as well as the use of IVFEs containing fish oil, are proven preventive options [29,84,106,107]. Since sepsis and infection have been related to liver impairment, the prevention and management of infection is also recommended. Risk factors and interventions to reduce the risk of development of PNALD are described in Table 4.

Practical recommendations/key points:Liver impairment during PN is multifactorial. Most of the common causes, apart from PN, are drugs, procedures or sepsis, or pre-existing liver disease.Establish at least a minimal EN intake to minimize risk of biliary complications and help reduce liver function test values.Future research: Dose-dependent effect for IVLEs containing omega-3 fatty acids, as well as the role of other components such as alpha-tocopherol in hepatobiliary complications.

#### 5.1.5. Catheter-Related Complications

CRBSI is one of the most important complications associated with PN. Infections of the central line can lead to sepsis, shock, and death. CRBSIs are a risk factor to be considered when inserting and managing any central VAD as they can increase morbidity, mortality, length of stay, and costs [3,24]. PN is historically considered to be an additional risk factor for CRBSI [108]. Evidence-based guidelines for line care and placement should be implemented and must be followed to minimize the risk of infection [3,8,109]. Tools and strategies for preventing CRBSI have been developed; the major recommendations include (1) cognitive-science-based educating and training healthcare workers who insert and maintain VADs; (2) using a checklist to improve the adherence to hygiene protocols; (3) the use of ultrasound guidance in the insertion of VADs; (4) the use of an antiseptic barrier cap and needleless secure devices; (5) abandoning the convenience of multi-lumen catheters; and (6) the use of antimicrobial-antiseptic impregnated VADs [3,109,110,111].

Other catheter-related complications include damage to surrounding structures, bleeding, venous thrombosis and/or stenosis and line obstruction (due to precipitation of infusates and thrombosis). The risk of thrombosis can be reduced by using aseptic placement techniques, smaller-gauge catheters, ensuring the correct placement of the tip of the catheter, avoiding PICCs, and proper fixation techniques [112,113]. Precipitation can be prevented by appropriate nursing care with appropriate nursing protocols for maintenance of the line, including the use of pumps for PN infusion [3,109]. In addition, the use of in-line filters, as well as the analysis throughout the process of prescription, validation, and compounding of the PN, may also help to reduce the risk of precipitation.

Practical recommendations/key points:Emphasize the importance of educational programs regarding protection against infection and hand decontamination.Protocolize VAD care and surveillance.Future research: Assessment of antiseptics or new drugs to address the development of new strategies to reduce CRBSI.

### 5.2. Monitoring

Metabolic complications are more common in patients receiving PN than those receiving EN. Therefore, close monitoring is required for the prevention and early detection of complications. The frequency of monitoring and review depends on the status of the patient’s clinical stability. Renal function, liver tests, glycemia, and serum electrolyte and triglyceride levels should be checked daily until stable and then at least every week (more frequently in critically ill patients or patients at risk of RS). Efficient monitoring can result in reduced PN-associated complications and reduced costs. To minimize complications and ensure safety during PN, all hospitals should have PN guidelines and standardized procedures. These should include recommendations for the assessment and documentation of the indications for PN, the PN prescription and goals of treatment, and required clinical and laboratory monitoring, including the identification and management of all the complications related to PN, including RS [6,8,34].

Practical recommendations/key points:Special situation, such as high electrolyte imbalance or refeeding syndrome, may require more frequent and prolonged laboratory monitoring.

## 6. Disease-Specific PN

### 6.1. Acute Pancreatitis

According to the ESPEN guidelines, patients with AP should be considered at moderate to high nutritional risk because of the catabolic nature of the disease and because of the impact of nutritional status on disease development [114]. However, patients with predicted severe AP should always be considered at high nutritional risk. The focus of early nutritional therapy, therefore, should be on these patients, since they are more ill and are unable to tolerate oral intake for longer periods of time. Complications of severe AP, such as bowel obstruction, abdominal compartment syndrome, prolonged paralytic ileus and mesenteric ischemia, may occur and represent a contraindication for EN [115,116]. The treatment of AP has historically included restricting oral and enteral intake to decrease pancreatic secretion until the complete resolution of symptoms. Currently, numerous studies have shown that the initiation of EN in AP is safe and associated with a decreased length of hospitalization and adverse effects [117,118,119,120].

There are sufficient data from several randomized clinical trials (RCTs) and systematic reviews/meta-analyses supporting the fact that in patients with severe AP, EN is safe and well-tolerated, with a significant reduction in complication rates, length of hospital stays, multi-organ failure, and mortality [114,121,122,123,124,125,126]. Given these data, both the American Gastroenterological Association and ESPEN recommend that EN should be preferred to PN in patients with AP and an intolerance of oral feeding [114,127]. A more recent study has also explored this topic and reached the same conclusions [128]. PN is therefore only recommended in patients with AP who do not tolerate EN or who are unable to tolerate targeted nutritional requirements, or if contraindications for EN exist. Apart from the mentioned advantages, EN is less expensive compared to PN [129].

The ESPEN guidelines recommend the use of parenteral glutamine in PN, but not a role for immunonutrition in severe AP [114]. This recommendation is based on four metanalyses with results in favor of glutamine supplementation [130,131,132,133]. Moreover, a recent meta-analysis has also shown positive results regarding the use of parenteral glutamine in severe AP [134]. Regarding the use of omega-3 fatty acids, a meta-analysis has shown that the administration of omega-3 fatty acids is beneficial for reducing mortality, infectious complications, and length of hospital stay, especially when received parenterally. However, the sample size in the metanalysis was small; only three RCTs assessing parenteral omega-3 fatty acids were included for each outcome [135]. Given that AP is a risk factor for hyperglycemia and hypertriglyceridemia, the close monitoring of blood sugar and triglyceride levels is recommended.

Practical recommendations/key points:AP patients with PN should be closely monitored for hyperglycemia and hypertriglyceridemia.Future research: Using PN in AP, the role of nutrients such as omega-3 fatty acids in reducing complications should be assessed. We will continue by studying the dose and length of glutamine addition, and confirming the role of glutamine in AP, especially patients critically ill with AP. As with other clinical situations, an assessment of the optimal PN initiation when PN is indicated should be performed.

### 6.2. Surgery Patients

The impact of preoperative nutritional status on postoperative morbidity and mortality is well documented. In malnourished patients screened before surgery, a 7–10-day course of preoperative nutritional therapy is indicated. Multimodal interventions focused on postoperative recovery help to achieve the early recovery of gastrointestinal function after surgery, as well as reduce the need for PN. However, optimal early oral nutrition may not be possible in all patients, especially those undergoing major abdominal surgery.

The ESPEN guidelines on clinical nutrition in surgery recommend that if oral and enteral intake cannot cover >50% of requirements for more than 7 days, a combination of enteral nutrition and PN is recommended [5]. Recently, a RCT stated that early supplemental PN was associated with reduced nosocomial infections in abdominal surgical patients [136]. However, there was no difference in the secondary outcomes, including length of hospital stay and total adverse events. In addition, as described by Ljungqvist et al., further treatment details are needed to interpret these results, since the average length of hospital stay was longer than seen in Enhanced Recovery After Surgery (ERAS) programs [137]. Consequently, PN will be indicated in patients with prolonged recovery. Accordingly, PPN administration has been suggested as an ERAS intervention since it can facilitate the provision of timely nutrition support during the perioperative period in selected patients [47,49]. Accordingly, findings from small studies suggest that PPN is a feasible approach for providing nutritional support to selected patients in perioperative settings [138,139,140,141].

Indications for postoperative PN include surgical complications such as disordered motility, including prolonged paralytic ileus, but also leaks or anastomotic breakdown, chyle leaks, and high-output fistula [5]. PN is beneficial in undernourished surgical patients in whom EN is not feasible or not tolerated, such as in intestinal obstruction. Several meta-analyses examining the influence of PN on morbidity and mortality in surgery showed a lower complication rate in malnourished surgical patients receiving PN, with no effects on the mortality rate [142,143]. However, when comparing EN and PN, EN seems to have the best risk-benefit ratio. Compared with PN, early EN has fewer complications and reduces length of hospitalization and costs [5,144,145]. The meta-analysis conducted by Zhao et al. concluded that patients who received EN had a shortened length of stay than those who received PN after major abdominal surgery [144]. Recent metanalyses have also shown positive outcomes related to EN compared to PN after cystectomy and pancreaticoduodenectomy [146,147].

According to ESPEN guidelines, glutamine supplementation should be considered for surgical patients with PN [5]. A recent meta-analysis, including 19 RCTs for elective major abdominal surgery, showed that glutamine parenteral supplementation reduced the hospital stay but did not affect overall mortality [148]. However, more recently, a multicenter RCT with surgical ICU patients did not find differences in mortality rate between the glutamine-supplemented group and control group [149]. Clinical advantages, such as lower risk of infection, hospital and intensive care unit stays, and infections, were shown in a meta-analysis comparing PN enriched with omega-3 fatty acids with a standard IVFE [5,79]. Thus, in surgical patients with PN, IVFE containing omega-3 fatty acids offers clinical benefits over standard solutions without omega-3 fatty acids and its use is recommended whenever possible [5,106].

Practical recommendations/key points:Consider preoperative PN in severely malnourished patients who cannot tolerate oral or enteral intake.Future research: Investigate the role of nutrients added to PN, such as omega-3 fatty acids and glutamine, in clinical outcomes (time and dose). Optimal timing for PN initiation to achieve better outcomes.

### 6.3. Critically Ill Patients

In critically ill patients, the addition of PN or the use of full PN in the acute phase should be considered individually. In fact, every critically ill patient staying for more than 48 h in the ICU is considered at risk of malnutrition [9,10]. To avoid large cumulative energy and protein deficits, EN and PN may be combined [150,151,152,153]. However, the timing of the initiation of EN or PN or a combination of both remains controversial [16,154,155]. As previously mentioned, inappropriate use of PN in ICU patients leads to worse outcomes, as has been shown in the EPANIC study [16].

Differences in the ASPEN and ESPEN guideline recommendations regarding the amount of protein and energy, and the timing of PN (full or combined with EN) are shown in Table 5 [9,10,66]. Current ASPEN recommendations for starting PN differ from those of the previous ASPEN guidelines. The current recommendation is based on two large trials comparing early PN with EN [156,157]. However, the value of this recommendation might be low, since it is directed to patients who may be candidates for EN during the first week of ICU and EN should be preferred. Therefore, supplemental PN is suggested to achieve nutritional requirements. Nevertheless, the authors recommend not initiating supplemental PN prior to day 7 of ICU admission. Furthermore, a recent meta-analysis did not find significant effect of a combination of EN with PN versus EN on the analyzed endpoints (mortality, length of stay, ICU stay, and ventilation days), but it did find improvements in nutrition intake [158].

Although indirect calorimetry to measure energy requirements is recommended, weight-adapted formulas may be used instead. Given that overfeeding has been associated with poor outcomes in critically ill patients, both societies recommend avoiding it, particularly in the early stages of treatment. The ASPEN guidelines currently recommend a higher protein target compared to the ESPEN recommendations; however, the influence of protein on the outcome of critically ill patients has also been controversial [9,10,155]. A recent meta-analysis comparing high versus low protein intake did not find a significant influence on overall mortality or other clinical outcomes [159]. Therefore, the indication, timing and protein amounts administered with PN to the critically ill have become more critical and individualized. In addition, the risk of overfeeding might be increased by supplemental PN. Thus, ESPEN guidelines state “In patients who do not tolerate full dose EN during the first week in the ICU, the safety and benefits of initiating should be weighed on a case-by-case basis” [10].

In general, the use of parenteral glutamine is not recommended, especially in renal failure, meanwhile the ESPEN guidelines recommend the use of IVFEs enriched with omega-3 fatty acids in critically ill patients [9,10,66].

Practical recommendations/key points:Avoid overfeeding: Initiation of supplemental PN should only be considered in stable, critically ill patients with a clearly insufficient EN intake.Reassess malnutrition and nutrition plan, especially during a prolonged ICU stay.In patients on renal replacement therapy, an increased provision of amino acids and micronutrients (electrolytes, vitamins and trace elements) should be considered.Future research: Monitoring the risk of overfeeding and new approaches to measuring it. Easy and feasible energy-requirement assessment suitable for the phase of illness in critically ill patients. Optimal timing for PN initiation, and timing for increased caloric and protein intake should be also assessed.

### 6.4. Inflammatory Bowel Disease

Malnutrition can occur in IBD, mainly in the active disease and to a greater degree in Crohn’s disease (CD) rather than in ulcerative colitis (UC). This is because of the capacity of CD to affect any part of the gastrointestinal tract, unlike UC, which is restricted to the colon [160]. Malnourished patients with IBD are more likely to be hospitalized and be admitted to hospital due to infection [161]. Common indications for PN in IBD include an obstructed bowel where there is no possibility of a feeding tube being placed beyond the obstruction or where this has failed, as well as other complications such as an anastomotic leak or a high-output intestinal fistula [160,162]. The ESPEN guidelines for IBD particularly recommend that CD patients with a proximal fistula and/or a very high output fistula should receive nutritional support by partial or total PN. Since trials show no benefit of PN in the management of acute severe UC, both in terms of inflammatory disease and precolectomy optimization, it is only recommended in malnourished patients with UC and severe disease when EN cannot to be tolerated or oral nutrition or EN are not effective [160,163,164,165]. As previously mentioned, EN is recommended over PN for use in preoperative optimization [5,163,166]. Thus, PN should be initiated preoperatively in CD patients at risk of malnutrition who have contraindications to EN or who are not able to maintain >60% of nutritional requirements orally [160,162]. Despite insufficient data supporting the routine use of PPN in IBD, it has been suggested that PPN could be advantageous in certain malnourished patients, especially when central VAD insertion may be delayed for logistical reasons.

Since there is no good evidence that the daily protein needs of IBD patients in remission differ from that recommended for the general population, the provision of 1 g of protein for each kilogram of body weight is considered reasonable. Meanwhile, specific formulations or substrates (e.g., glutamine, omega-3-fatty acids) are not highly recommended with EN or PN in IBD patients [160,162].

Practical recommendations/key points:As malnutrition is highly prevalent in IBD patients, screening for malnutrition should be implemented for all such patients.Per the previous point, before nutritional support is initiated, an adequate nutritional plan (amount of macro and micronutrients) should be established to avoid RS and electrolyte imbalances.Future research: Comparing outcomes of the role of supplemental PN when EN does not achieve optimal requirements.

## 7. Conclusions

In recent decades, significant advances in PN formulations and processes have led to an enhanced understanding of what is safe, thereby reducing PN complications. PN is a specialist therapeutic intervention that can help to optimize the nutritional status in hospitalized patients. Appropriate use of PN improves nutritional parameters and reduces the risk of complications associated with malnutrition. However, inappropriately initiated PN is associated with unnecessary risk to the patient and increased costs. Therefore, the implementation of programs that assess and use methods and procedures to reduce complications related to PN, and improve clinical outcomes, should be prioritized.

## Figures and Tables

**Table 1 nutrients-14-04480-t001:** Examples of clinical conditions requiring PN [4,7,8].

Condition	Mechanism/Indication for PN	Example
Short bowel Intestinal fistula Extensive intestinal mucosal disease	Reduction of absorption capacity Loss of nutrients	Short bowel syndrome, ischemic bowel, complications of colorectal or bariatric surgery, high-output stoma, high-output intestinal fistula Radiation or chemotherapy-related enteritis, mucositis, autoimmune enteropathy, gut graft-versus-host disease
Mechanical bowel obstruction	Blockage of intestinal lumen Recurrent vomiting	Malignant bowel obstruction, intestinal adhesions, stenosis or strictures, inflammatory disease, peritoneal carcinomatosis
Motility disorders	Failure to tolerate adequate oral or enteral intake Recurrent vomiting	Functional gastrointestinal disorders, ileus, scleroderma, acute pancreatitis, post-operatively, gastrointestinal failure associated with critical illness, pseudo-obstruction, adhesive disease
Bowel rest needed	Need to restrict oral or enteral intake	Ischemic bowel, perioperative status, acute pancreatitis, chylous fistula
Other	Failure of oral or enteral nutrition	Unable to achieve or maintain secure oral or enteral access

**Table 2 nutrients-14-04480-t002:** Types of VAD used for PN administration [3,8,24,26].

Type of VAD	Placement	Limitations	Advantages
Short peripheral catheter	Percutaneous peripheral insertion.	Infusion < 600 mOsm/L, high risk of phlebitis.	Easy to place, cost, lower infection risk.
Midline	Percutaneous peripheral insertion.	Not appropriate for infusions > 900 mOsm/L (needs central access).	Lasting 2–4 weeks.
PICC	Percutaneous placement via a peripheral vein (basilic, cephalic or brachial vein).	Self-care difficult, uncomfortable for long periods, placement needs trained personnel.	Low risk of placement complications. Used in acute and home care settings. Easy to remove. Lasting weeks to months.
Nontunneled central VAD	Subclavian, jugular or femoral vein.	Operating room or hospital setting for placement.	Long-term usage, easy self-care.
Tunneled central VAD	Subclavian or jugular (Hickman, Broviac, Hohn types).	Hospital setting, small procedure for removal.	Lower risk of infection, position on chest facilitates self-care; lasting months to years (home PN).
Implanted ports	Subclavian or jugular.	Hospital setting, surgical procedure for removal, needle access required.	Associated with lower risk of infection.

**Table 3 nutrients-14-04480-t003:** Characteristics of IVFEs for PN [74,84,85].

Lipid Source and Content	Phytosterol Content (mcg/mL)	Commercial Name (Manufacturer)
Soybean oil 100%	422–439	Intralipid^®^ 20% (Fresenius Kabi, Bad Homburg, Germany)
Soybean oil 50% Coconut oil 50%	187–278	Lipofundin^®^ 20% (BBraun, Melsungen, Germany)
Soybean oil 64% Coconut oil 36%	346	Structolipid^®^ 20% (Fresenius Kabi, Germany)
Soybean oil 20% Olive oil 80%	208–274	ClinOleic^®^/ClinoLipid^®^ 20% (Baxter Deerfield, USA)
Fish oil 100%	0	Omegaven^®^ 10% (Fresenius Kabi, Germany)
Soybean oil 40% Coconut oil 50% Fish oil 10%	140	Lipiderm^®^/LipoPlus^®^ 20% (BBraun, Germany)
Soybean oil 30% Coconut oil 30% Olive oil 25% Fish oil 15%	124–207	SMOFlipid^®^ 20% (Fresenius Kabi, Germany)

**Table 4 nutrients-14-04480-t004:** Risk factors and interventions to reduce risk for development of PNALD.

Type of Risk Factor	Cause	Reason	Intervention
Unrelated to PN	Sepsis and/or insult	Liver toxicity	Infection prevention
	Drugs-induced toxicity	Drugs causing liver toxicity	Identify the drug, change if possible
Related to PN	Lack of enteral intake	Impaired secretion of bile or biliary obstruction	Trophic EN, reintroducing enteral/oral intake
	Overfeeding	Fat accumulation leading to steatosis	Reduce total energy intake (fat and/or glucose) or change to enriched fish-oil IVFE
	Lipids with a high phytosterol load	Phytosterols direct/indirect action in the liver	Change to lipid with lower phytosterol content and/or reduce lipids

**Table 5 nutrients-14-04480-t005:** ASPEN and ESPEN recommendations for critically ill patients with PN.

Society	Start SPN *	Start PN	Protein g/kg/day	Energy Kcal/kg/day
ASPEN	After 6 days	Any time	1.2–2.0	12–25 (up to 7–10 day)
ESPEN	Within 3–7 days	Within 3–7 days	1.3	Not exceeding 70% of EE ** (day 1–3)After day 3: 80–100% EE **

* SPN, supplemental PN; ** EE, estimated energy, calculated by indirect calorimetry.

## Data Availability

Not applicable.

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
