# Peer review of "Parenteral Nutrition Overview"

_nutrients, 2022, doi:10.3390/nu14214480_

Round 1
Reviewer 1 Report (New Reviewer)
To the editor,
This article, even if it is very well documented and referenced, in various aspects of parenteral nutrition appears very low in terms of methodology to be considered as a review of literacy on this topic.
Above all, the strategy for searching relevant references and the construction of the manuscript is not defined. The quality of the references is not discussed. Objective is not enough clearly defined at the beginning. The manuscript does not rely sufficiently on the citation of recommendations from American and European societies of Nutrition whenever possible.
The aspect about recommended formulation and the relevant references is finally scarce.
A large part part about compounding is lacking. Various sentences are lacking or potentially wrong.
Large part about peripheral PN is given even if in practice its use should be discouraged in a majority of situation.
I am not an expert for every parts of this manuscript. So I only give some minor remarks on certain parts.
The abstract is not representative of the general manuscript.
P1 : line 9 also : pharmaceutical fields
L14 : “can be” has to be replaced by « has been »
L72 : indication : Failure to administer oral or enteral nutrition : need to minor sentence or to adjust the sentence and give reference (from recommendations) if available
L77 : « Before initiating PN, a full assessment of the feasibility of EN should be conducted, so that PN is reserved for clinical conditions where adequate EN is not an option.”
It should include the ability to associate PN and EN together.
L 139 : “Thus, it is reasonable to limit peripheral PN < 900 mOsm/L to pediatric patients and < 800-850 mOsm/L to adults”.
I would not recommend to add this sentence because there is no reference to support this sentence, and I would be kept in head that it is not recommended to use peripheral route, and it would depend on the time this access is used.
P144 “Although the femoral vein is often used”,
I disagree : maybe used in critical care, but for HPN this access is rarely used
L159, L197, L544 : do not use term “solution” because it is an emulsion
L172 : some countries in Europe : there are recommendations from European societies of Nutrition ; it is not the same level of recommendation for children and adults and it would be helpful to add this
178 : administration of lipids in 24 h usually
L196 : replace role by interest
PPN is not defined
L234 : HPCB should consider also compounding of stocked bags in hospital (standards) ; there is a large literacy on this topic.
L235 : because of limited mixture stability : no, bags are compounded every day because of provision of individualized bags to correspond to individual needs, for critical patients, first of all.
L238 : more than 24 months
L268 : ‘Bicarbonate salts should be avoided due to incompatibility with PN admixtures.’
Bicarbonate is reputated to be incompatible with various drugs because a majority of API are more stable in acidic condition ; it would raise the opportunity to use lactate ou acetate to be less acidic in case of hyperchloremia
L275 :
Temperature: for example, with increased temperature, there is a raised likelihood of precipitate formation and degradation of several amino acids.
It is often not true, except for Calcium and phosphate
L 277 : pH: IVFEs are more stable at higher pH values,
It should give more accurate reference and globally IVFE are stable for acidic and neutral pH values. A literacy about formation of oxidative degradation with dextrose according to pH is not given.
IVLE : not defined
L304 : For stability, avoid high loads of electrolytes in the PN admixture.
The sentence appears naïve.
L306 : Evaluate differences in outcomes between PPN and central PN, including cost-effectiveness analysis
I disagree because the first level of analysis should be based on clinical argument.
L584 : “various recommendation for crbsi” ; management and prevention tools are available
L589 : for reducing precipitation, a high level of analysis of PN composition should help ; and use of in line filters should be helpful.
Author Response
I thank the reviewer for your time spent carefully reviewing the manuscript and your opinions for providing valuable feedback on my work. In what follows the reviewers’ comments are in black and the author’s responses are in red.

Reviewer 2 Report (New Reviewer)
Summary: Thank you for contributing this excellent manuscript on the overview of parenteral nutrition. Very well written and touches on the important aspects of PN. No major critiques from my standpoint. Good job on pointing out where differences may be present in different societies or countries.
Specific comments:
Line 46: Switch “chapter” to “section”
Line 48: Remove “The”
Line 70: Consider rewording sentence to “critically ill, surgical, and/or cancer patients”.
Line 84: Add “initiation” after PN so it reads “The timing of PN initation…” to make this more clear what the timing is referring to.
Line 124: Consider rephrasing sentence to say that patients on fluid restriction should not be given PPN due to high volume.
Line 144: I would consider saying the femoral vein “can be used” rather than “often used” because I think most people use that as a last resort. I feel like the sentence reads currently like that is the first choice for a site for providers.
Line 234: What about infusion companies that compound formulas for HPN patients? They get weekly shipment. May consider including that as well to cover all bases.
Line 419: I would adjust order of the complications to match the order they are discussed in the section: metabolic, infectious, and mechanical.
Author Response
I thank the reviewer for your time spent carefully reviewing the manuscript and your opinions for providing valuable feedback on my work. In what follows the reviewers’ comments are in black and the author’s responses are in red.

This manuscript is a resubmission of an earlier submission. The following is a list of the peer review reports and author responses from that submission.
Round 1
Author Response
I appreciate you and your time in reviewing my manuscript and providing valuable comments. It was your valuable and insightful comments that led to possible improvements in the current version. I have carefully considered the comments and tried my best to address every one of them. Attached I provide the point-by-point responses.

Reviewer 2 Report
1. Line 191, ILE should be spelled out for the first time.
2. This paper is good at reviewing the basic knowledge of parenteral nutrition. However, most of information can be found in books and guidelines. The author needs to provide some novel opinion or identify gaps in existing studies for potential future research.
3. The compatibility and stability of additives, including electrolytes, trace elements, and vitamins, to parenteral nutrition should be reviewed. Also, the storage to avoid photodegradation, the interaction and precipitation (e.g., particle size) of nutrients and the neonatal use of PN need to be reviewed.
4. The concept, composition and applicability of TPN and PPN are different and need to be reviewed separately. For example, PPN instead of TPN is recommended in patients with acute pancreatitis if they are malnourished. The applications of TPN and PPN are also different in surgical, critically ill, and IBD patients.
5. The manuscript has many grammar mistakes, typos, and non-English writing styles which needs to be corrected. Please seek assistance from a competent native English.
Author Response

(The authors gave the same response as above.)
